# Understanding High-Dimensional Bayesian Optimization

**Leonard Papenmeier** [1]  **Matthias Poloczek** [2]  **Luigi Nardi** [1][3]

## Abstract

Recent work reported that simple Bayesian optimization (BO) methods perform well for high-dimensional real-world tasks, seemingly contradicting prior work and tribal knowledge. This paper investigates why. We identify underlying challenges that arise in high-dimensional BO and explain why recent methods succeed. Our empirical analysis shows that vanishing gradients caused by Gaussian process (GP) initialization schemes play a major role in the failures of high-dimensional Bayesian optimization (HDBO) and that methods that promote local search behaviors are better suited for the task. We find that maximum likelihood estimation (MLE) of GP length scales suffices for state-of-the-art performance. Based on this, we propose a simple variant of MLE called MSR that leverages these findings to achieve state-of-the-art performance on a comprehensive set of real-world applications. We present targeted experiments to illustrate and confirm our findings.

## 1. Introduction

Bayesian optimization (BO) has found wide-spread adoption for optimizing expensive-to-evaluate black-box functions that appear in aerospace engineering (Lukaczyk et al., 2014; Lam et al., 2018), drug discovery (Negoescu et al., 2011), robotics (Lizotte et al., 2007; Calandra et al., 2016; Rai et al., 2018; Mayr et al., 2022) or finance (Baudiš & Pošík, 2014).

While BO has proven reliable in low-dimensional settings, high-dimensional spaces are challenging due to the curse of dimensionality (COD) that demands exponentially more data points to maintain the same precision with increasing problem dimensionality. Several approaches have extended BO to high-dimensional spaces under additional

assumptions on the objective function that lower the data demand, such as additivity (Duvenaud et al., 2011; Kandasamy et al., 2015; Hoang et al., 2018; Han et al., 2021; Ziomek & Ammar, 2023; Bardou et al., 2024) or the existence of a low-dimensional active subspace (Wang et al., 2016; Nayebi et al., 2019; Letham et al., 2020; Papenmeier et al., 2022). Without such assumptions, it was widely believed that BO with a Gaussian process (GP) surrogate is limited to approximately 20 dimensions for common evaluation budgets (Frazier, 2018; Moriconi et al., 2020). Recently, Hvarfner et al. (2024) and Xu & Zhe (2024) reported that simple BO methods perform well on high-dimensional real-world benchmarks, often surpassing the performance of more sophisticated algorithms.

Due to its many impactful applications, high-dimensional Bayesian optimization (HDBO) has seen active research in recent years (Binois & Wycoff, 2022; Papenmeier et al., 2023). While the boundaries have been pushed significantly, the causes of performance gains have not always been thoroughly identified. For example, in the case of the `BODi` (Deshwal et al., 2023) and `COMBO` (Oh et al., 2019) algorithms, later work found that the methods benefited from specific benchmark structures prevalent in their evaluation (Papenmeier et al., 2023). Similarly, an evaluation of Nayebi et al. (2019) showed that the performance of some prior methods is sensitive to the location of the optimum in the search space.

This paper is motivated by the recent call for more scrutiny and exploratory research (Herrmann et al., 2024). By exhaustive experimentation, we first identify underlying challenges arising in high dimensions and then examine state-of-the-art HDBO methods to understand how they mitigate these obstacles. Equipped with these insights, we propose a simpler approach that uses maximum likelihood estimation (MLE) of the GP length scales called MLE Scaled with RAASP (MSR). We demonstrate that MSR is sufficient for state-of-the-art HDBO performance without the need for specifying a prior belief on length scales as in maximum a-posteriori estimation (MAP). We note that practitioners usually do not possess such priors and instead rely on empirical performances on benchmarks. In particular, we change the initialization of length scales to avoid vanishing gradients of the GP likelihood function that easily occur in high-dimensional spaces but, so far, have been overlooked for BO.

[1]Department of Computer Science, Lund University, Lund, Sweden [2]Amazon (This research does not relate to Matthias' work at Amazon.) [3]DBtune. Correspondence to: Leonard Papenmeier <leonard.papenmeier@cs.lth.se>.

*Proceedings of the $42^{nd}$ International Conference on Machine Learning*, Vancouver, Canada. PMLR 267, 2025. Copyright 2025 by the author(s).

Furthermore, we provide empirical evidence suggesting that good BO performance on extremely high-dimensional problems (on the order of 1000 dimensions) is due to local search behavior and not to a well-fit surrogate model. In summary, we make the following contributions.

1. We identify underlying challenges that arise in HDBO and explain why recent methods succeed. We show that vanishing gradients and local search behaviors are important in HDBO.

2. We find that MLE of GP length scales suffices for state-of-the-art performance. We propose a simple variant of MLE called `MSR`.

3. We evaluate `MSR` on a comprehensive set of real-world applications and a series of targeted experiments that illustrate and confirm our findings.

## 2. Problem Statement and Related Work

We aim to find $\boldsymbol{x}^* \in \arg\min_{\boldsymbol{x} \in \mathcal{X}} f(\boldsymbol{x})$, where $f : \mathcal{X} \to \mathbb{R}$ is an unknown, only point-wise observable, and expensive-to-evaluate black-box function and $\mathcal{X} = [0,1]^d$ is the $d$-dimensional search space, sometimes called "input space". Bayesian optimization (BO) is a popular approach to optimize problems with the above characteristics. We give a summary of BO and Gaussian processes (GPs) in Appendix A and restrict the discussion to high-dimensional BO.

Extending the scope of BO to high-dimensional problems was, for a long time, considered "as one of the holy grails" (Wang et al., 2016) or "one of the most important goals" (Nayebi et al., 2019) of the field. Several contributions extended the scope of BO to specific high-dimensional problems. However, for the longest time, no fully scalable method has been found to extend in arbitrarily high-dimensional spaces without making additional assumptions about the problem structure. The root problem of extending Bayesian optimization (BO) to high dimensions is the curse of dimensionality (COD) (Binois & Wycoff, 2022) that not only requires exponentially many more data points to model $f$ with the same precision but also complicates the fitting of the GP hyperparameters and the maximization of the acquisition function (AF). The growing demand for training samples stems from increasing point distances in high dimensions, where the average distance in a $d$-dimensional hypercube is $\sqrt{d}$ (Köppen, 2000).

This paper focuses on high-dimensional Bayesian optimization (HDBO) operating directly in the search space. Other methods for HDBO include linear (Nayebi et al., 2019; Wang et al., 2016; Letham et al., 2020; Papenmeier et al., 2022; 2023) or non-linear (Tripp et al., 2020; Moriconi et al., 2020; Maus et al., 2022; Bouhlel et al., 2018; Chen et al., 2020) embeddings to map from a low-dimensional subspace

to the input space.

**High-dimensional BO in the input space.** In the literature, HDBO operating directly in the high-dimensional search space $\mathcal{X}$ is often considered infeasible due to the COD. Numerous approaches have been proposed, often leveraging assumptions made on the objective function $f$ such as additivity (Kandasamy et al., 2015; Gardner et al., 2017; Wang et al., 2018; Mutny & Krause, 2018; Ziomek & Ammar, 2023) or axis-alignment (Eriksson & Jankowiak, 2021; Hellsten et al., 2023; Song et al., 2022), which simplify the problem and improve sample efficiency if they are met. Other methods identify regions in the search space relevant for the optimization, for example, using trust regions (TRs) (Regis, 2016; Pedrielli & Ng, 2016; Eriksson et al., 2019) or by partitioning the space (Wang et al., 2020).

Recently, multiple works re-evaluated basic BO setups for high-dimensional problems, presenting state-of-the-art performance on various high-dimensional benchmarks with only small changes to basic BO strategies. Hvarfner et al. (2024) use a dimensionality-scaled log-normal length scale hyperprior that shifts the mode and mean of the log-normal distribution by a factor of $\sqrt{d}$, designed to counteract the increased distance between randomly sampled points. To optimize the AF, they change `BoTorch`'s (Balandat et al., 2020) default strategy of performing Boltzmann sampling on a set of quasi-randomly generated points by sampling over both a set of quasi-randomly generated points and a set of points that are generated by perturbing the 5% best-performing points. By perturbing 20 dimensions on average, this strategy creates candidates closer to the incumbent observations and enforces a more exploitative behavior (Regis & Shoemaker, 2013; Regis, 2016; Eriksson et al., 2019). The effect of the sampling strategy was recently revisited by Rashidi et al. (2024). They argue that `TuRBO`'s random axis-aligned subspace perturbations (RAASPs) are crucial to performance on high-dimensional benchmarks and, motivated by this observation, derive the cylindrical Thompson sampling (TS) strategy that maintains locality but drops the requirement of axis alignment. Independently of Hvarfner et al. (2024), Xu & Zhe (2024) reported that "standard GPs can be excellent for HDBO" which they show empirically on several high-dimensional benchmarks[1]. They use a uniform $\mathcal{U}(10^{-3}, 30)$ length scale hyperprior, which, in their experiments, performs superior to `BoTorch`'s Gamma $\Gamma(3,6)$ length scale hyperprior.

---

[1]We discuss an earlier preprint (https://arxiv.org/abs/2402.02746v3). In a later version, presented at ICLR 2025, the authors – concurrently to our work – developed an initialization strategy similar to the one presented in Section 4.

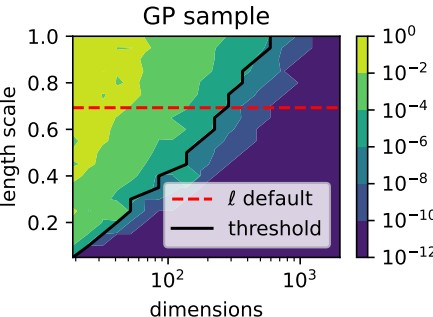

*Figure 1.* Maximum MLE gradient magnitude for the 50 first gradient steps initialized with different initial length scales ($y$-axis) and problem dimensionalities ($x$-axis). With short initial length scales, the gradients vanish even for low dimensions.

## 3. Facets of the Curse of Dimensionality

This section discusses how the curse of dimensionality impacts high-dimensional Bayesian optimization (HDBO) and techniques to mitigate these challenges.

### 3.1. Vanishing Gradients at Model Fitting

Bayesian optimization uses a probabilistic surrogate model of $f$ to guide the optimization. Gaussian processes (GPs) are the most popular surrogates due to their analytical tractability. They allow for different likelihoods, mean, and covariance functions, each often exposing several hyperparameters, including the function variance $\sigma_f^2$, the noise variance $\sigma_n^2$, and the $d$ model length scales $\ell$ that need to be fitted to the task at hand. In the absence of prior information about the objective function $f$, maximum likelihood estimation (MLE) is commonly used to fit the model hyperparameters by maximizing the GP marginal log-likelihood (MLL):

$$\boldsymbol{\theta}_{\mathrm{MLE}}^* = \arg\max_{\boldsymbol{\theta}} \log p(\boldsymbol{y}|X, \boldsymbol{\theta}), \qquad (1)$$

Here, $X$ are points in the search space $\mathcal{X}$, $\boldsymbol{y}$ are the associated function values, and $\boldsymbol{\theta}$ is the vector of GP hyperparameters. See Appendix A for more information.

The MLL is usually maximized using a multi-start gradient descent (GD) approach. A crucial component of fitting a GP is choosing starting points for the MLE hyperparameters. In this section, we show that an ill-suited length scale initialization scheme can cause the gradient of the MLL function with respect to the GP length scales to vanish for high-dimensional problems. Thus, the length scales remain at the numerical values that they have been initialized to and will not be fitted to the objective function.

Fig. 1 shows the severity of the vanishing-gradients phenomenon. We plot the maximum magnitude (element-wise) of the length scale gradient across 50 gradient updates of an isotropic GP with a $^5/_2$-Matérn kernel as a function of the input dimensionality of the objective function ($x$-axis) and

the initial value for the length scale hyperparameter with which the gradient-based optimizer starts when maximizing the MLE ($y$-axis). The objective function is sampled from a GP with a $^5/_2$-Matérn kernel and $\ell = 0.5$, i.e., a GP prior sample. We provide additional implementation details in Appendix B. The dashed line shows the default initial length scale of $\ln 2$ used in `GPyTorch`. We consider gradients smaller than the machine precision for single floating point numbers 'vanished'. The reason is that even after 500 gradient updates, the length scale would change at most by $\approx 6 \times 10^{-5}$ from the value with which the gradient-based optimizer was initialized.

**Methods to Mitigate Vanishing Gradients.** One strategy to counteract the vanishing gradients is to replace MLE with maximum a-posteriori estimation (MAP) by choosing a hyperprior on the length scales that prefers long length scales:

$$\boldsymbol{\theta}_{\mathrm{MAP}}^* = \arg\max_{\boldsymbol{\theta}} \underbrace{\log p(\boldsymbol{y}|X, \boldsymbol{\theta})}_{\text{evidence}} + \underbrace{\log p(\boldsymbol{\theta})}_{\text{prior}}. \qquad (2)$$

MAP maximizes the unnormalized log posterior, which is the sum of the MLL and the log prior. We sometimes use the terms 'MLL' and 'unnormalized log-posterior' interchangeably if what is meant is clear from the context. The gradient of the recently popularized dimensionality-scaled log-normal hyperprior of (Hvarfner et al., 2024) directs the optimizer toward the mode of the hyperprior $\log p(\boldsymbol{\theta})$ that corresponds to long length scales.

If the gradient-based optimizer of the MLE reaches sufficiently long length scales, the MLL $\log p(\boldsymbol{y}|X, \boldsymbol{\theta})$ no longer vanishes. Fig. 2 shows the length scales of a GP conditioned on random observations of a realization drawn from an isotropic 1000-dimensional GP prior with length scale $\ell = 0.5$ and a $^5/_2$-Matérn kernel when fitting with MLE and MAP. We initialize the length scales with $\ln 2 \approx 0.69$, draw 1 and 10 observations uniformly at random, and maximize the MLL for 500 iterations with MLE and MAP, using a log-normal hyperprior. We average over 10 random restarts. When conditioning on only 1 observation, the MAP optimization starts at the initialization $\ln 2$ and converges to the hyperprior mode. For 10 observations, the length scales first move toward the mode but then converge to a different point, which trades off the attraction of the prior mode and the ground truth length scale. With MLE, the length scale does not change because of the vanishing gradients issue.

To "ensure meaningful correlation" in increasingly high-dimensional space, Hvarfner et al. (2024) scale a log-normal hyperprior for the GP length scales with the problem's dimensionality. Xu & Zhe (2024) pursue a different approach of initializing the gradient-based optimizer of the length scales with large values. Specifically, they posit a uniform $\mathcal{U}(10^{-3}, 30)$ hyperprior to the length scales and initialize

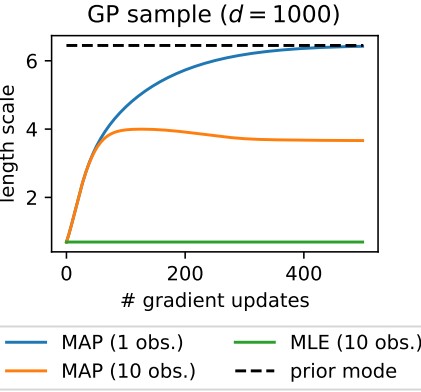

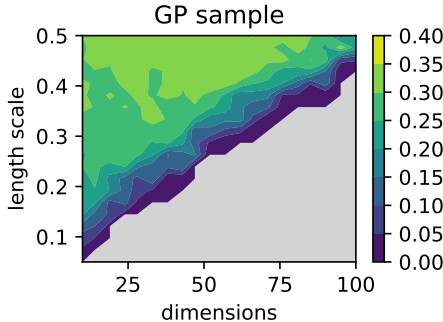

*Figure 3.* Average distances between the initial and the final candidates of `LogEI` for various model length scales and dimensionalities without RAASP sampling. Values in the gray region are numerically zero. In high dimensions, the gradient of the AF vanishes, causing no movement of the gradient-based optimizer.

*Figure 2.* Gradient-based optimization of a GP length scale with MAP and MLE. When conditioning the GP on only one random observation, MAP converges to the prior mode. We plot $\pm$ one standard error, which is too small to be visible.

the gradient-based optimization of the automated relevance determination (ARD) kernel with $d$ samples from the hyperprior. Although their method does not scale the hyperprior mode with the problem's dimensionality, the expected length scale of $\approx 15$ is sufficiently long to avoid vanishing gradients for the problems studied by the authors.

While these methods mitigate the issue of vanishing gradients, increasing length scales by any constant factor does not always solve it, as Fig. 1 indicates. Whether scaling the hyperprior of the length scale with the problem's dimensionality achieves a good fit of the surrogate model depends on the properties of $f$. The success of the two methods described above is related to the effect of the increased length scales on mitigating the problem of vanishing gradients. If the underlying function varies quickly, the GP needs to use a short length scale to model $f$. In such cases, the GP cannot model the function globally, as shown in Appendix C.4.

### 3.2. Vanishing Gradients of the Acquisition Function

Several popular acquisition functions for Bayesian optimization (BO), such as upper confidence bound (UCB) (Srinivas et al., 2010), expected improvement (EI) (Jones et al., 1998), and probability of improvement (PI) (Jones, 2001), rely solely on the GP posterior, exhibiting only small variation when the posterior itself changes only moderately. In high-dimensional spaces, the expected distance between two points sampled uniformly at random increases with $\sqrt{d}$ (Köppen, 2000). Thus, there are typically large 'unexplored regions' in the search space where the algorithm has not sampled. Suppose a commonly used GP with constant prior mean function and a stationary kernel. The GP posterior corresponds to the prior in those regions unless the kernel has sufficiently long length scales. Therefore, the GP posterior and the acquisition surface are flat in those vast unexplored regions. Acquisition functions (AFs) are usually

optimized with gradient-based approaches. Thus, these 'flat' areas of the AF also lead to vanishing gradients, but this time of the AF. This affects the selection of the next sample in the BO procedure.

The `LogEI` (Ament et al., 2024) AF provides a numerically more stable implementation of EI, mitigating the problem of vanishing gradients but not solving it, as we demonstrate next. Fig. 3 shows the average distances the gradient-based optimizer travels for `LogEI`. The surrogate is a GP with 20 observations drawn uniformly at random. The unknown objective function is a realization of the same GP. We run `BoTorch`'s multi-start GD acquisition function optimizer, initialized with 512 random samples and 5 random restarts, and measure the distances between the starting and end points of the gradient-based optimization of the AF. Average distances increase with $\sqrt{d}$; thus the plot shows average distances between the starting and endpoints of the gradient-based optimization that are normalized by $d^{-\frac{1}{2}}$. Gray regions correspond to a numerically zero average distance, indicating vanishing gradients. Vanishing gradients of the AF remain a problem, even when using `LogEI` and operating in moderate dimensions.

**Methods to Mitigate Vanishing Gradients of the AF.** One technique for handling vanishing gradients in the AF optimization is *locality*. `TuRBO` (Eriksson et al., 2019), for example, uses trust regions (TRs) to constrain the optimization to a subregion of the search space. Even if the GP surrogate is uninformative, `TuRBO` performs local search and can optimize high-dimensional problems even if data is scarce. `TuRBO` also uses random axis-aligned subspace perturbation (RAASP) sampling (Rashidi et al., 2024; Regis & Shoemaker, 2013), i.e., with a probability of $\min\left(1, \frac{20}{d}\right)$, it replaces the value of each dimension of the incumbent solution with a value drawn uniformly at random within the TR bounds. This process is repeated multiple times to

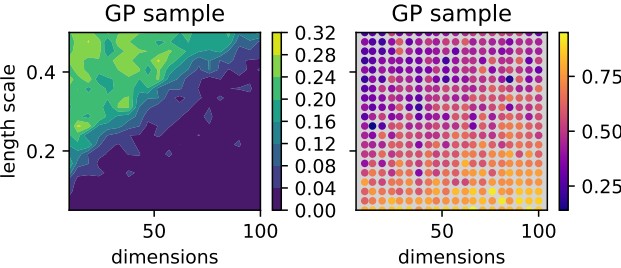

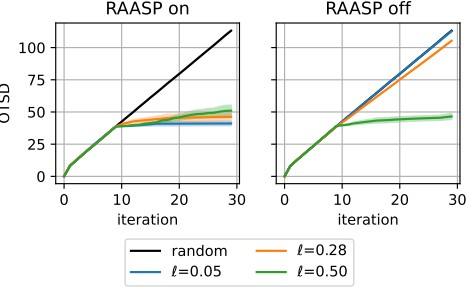

*Figure 4.* **Left**: Average distances between the initial and the final candidates of `LogEI` with RAASP sampling. The vanishing gradient issue decreases. **Right:** Fraction of multi-start GD candidates originating from the RAASP samples when evaluating `LogEI` on random samples. In high dimensions, RAASP samples are increasingly more likely to get picked, even for longer length scales.

*Figure 5.* OTSD for `BoTorch`'s AF maximizer operating on a 100-dimensional space and GPs of various length scales with and without RAASP sampling. The behavior of short-length-scale GPs reverts to local search ($\ell = 0.05$ in the left panel) with RAASP sampling and to local search without RAASP sampling ($\ell = 0.05$ and $\ell = 0.28$ in the right panel). Shaded areas show the standard error of the mean obtained by 10 random repetitions. In the right panel, the blue line masks the black line.

create several candidates evaluated on a realization of the GP posterior, a process known as Thompson sampling. The point of maximum value is then chosen for evaluation in the next iteration of the BO loop. This design choice further enforces locality as new candidates only differ on average from the incumbent in 20 dimensions for $d \geq 20$.

Differing slightly from `TuRBO`'s approach, RAASP sampling has been implemented in `BoTorch`'s AF maximization and can optionally be enabled with the parameter `sample_around_best`. `BoTorch` augments a set of globally sampled candidates with the RAASP samples, resulting in twice as many initial candidates. It perturbs the best-performing points by replacing the value of each dimension with a probability of $\min\left(1, \frac{20}{d}\right)$ by a value drawn from a Gaussian distribution, truncated to stay within the bounds of the search space. `BoTorch` then chooses the points of maximum initial acquisition value to start the GD optimization of the AF.

With increasing dimensionality or descending length scale, the starting points for the multi-start GD routine chosen by the AF maximizer are increasingly more likely to originate from the RAASP samples. Fig. 4 (right panel) illustrates this. Here, we draw realizations from GPs, initialized with different dimensionalities ($x$-axis) and length scales ($y$-axis). For each realization, we maximize the AF with RAASP sampling and plot the percentage of candidates of maximum acquisition value originating from the RAASP samples across all candidates. A higher percentage indicates a more 'local' sampling. We further average across five random restarts. The percentage of RAASP candidates with maximum acquisition value increases with the input dimensionality and decreases with the length scale. At the same time, those candidates stay close to the initial candidates. This is shown in the bottom right of Fig. 4 (left panel), which shows the average distance traveled by candidate points of the AF maximizer when using RAASP sampling. With RAASP sampling, candidates travel a positive distance, visualized

by the lack of gray color. This indicates a reduction of the vanishing-gradient issues. We attribute this to candidates being created close to the incumbent observations where the AF is less likely to be flat.

In general, when the length scales of the GP are short and the dimensionality is high, BO shows a local-search-like behavior with RAASP sampling and a random search behavior without it. We demonstrate this using the observation traveling salesperson distance (OTSD) (Papenmeier et al., 2025), which quantifies an algorithm's exploration by finding the shortest path connecting all observations made up to a certain iteration. The OTSD curve of an algorithm $A$ consistently lying above the curve of algorithm $B$ indicates that algorithm $A$ is more explorative as its observations are spread more evenly across $\mathcal{X}$. The OTSD is always monotonic increasing. Fig. 5 shows the OTSDs for 100-dimensional GPs with different length scales, each initialized with 10 random samples in the design of experiments (DOE) phase and subsequently optimized with `LogEI` and RAASP sampling for 20 iterations. Unless the model length scale is sufficiently long for the AF gradient not to vanish (as for $\ell = 0.5$ in the right panel of Fig. 5), the AF maximizer picks one of the initial random candidates without further optimizing it. This is supported by the trajectories for the BO phase (iteration $\geq 9$) following the trajectory of the DOE phase (iteration $< 9$) for $\ell = 0.05$ and $\ell = 0.28$ in the right panel of Fig. 5. We generally recommend the RAASP sampling method as it improves BO by automatically reverting to local search when encountering flat AFs.

### 3.3. Bias-Variance Trade-off for fitting the GP

GP models are commonly fitted by maximizing the MLL, either using unbiased MLE estimation or using MAP, which places a hyperprior on one or several GP hyperparameters.

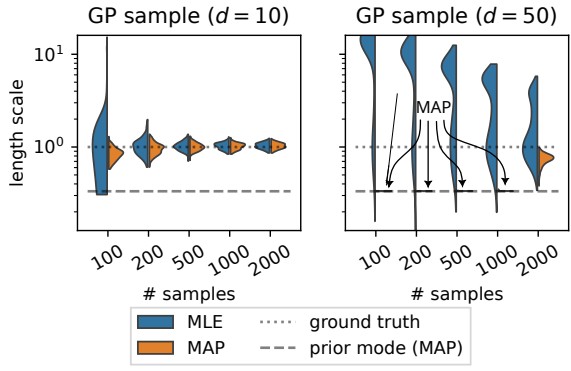

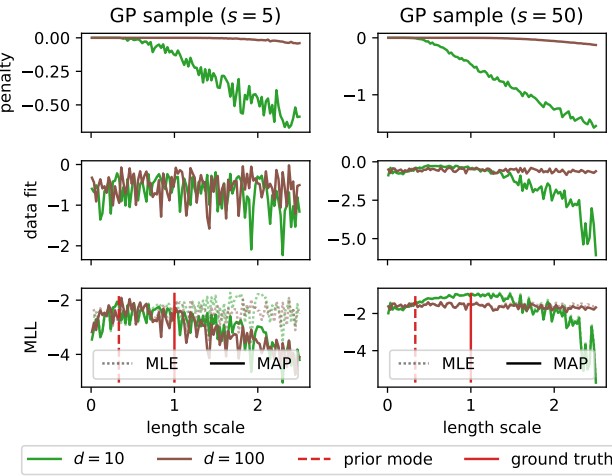

*Figure 6.* Average length scales ($y$-axis) obtained by MLE (blue) and MAP (orange) for different numbers of randomly sampled observations ($x$-axis) for a 10- and for a 50-dimensional GP prior sample. The obtained length scales differ substantially for the higher dimensional function if few points have been observed.

*Figure 7.* The MLL surface (bottom), the penalty (top row), and data fit terms (center) for various length scales and numbers of observations. Fewer observations lead to more erratic changes in the data fit term, leading to higher variance in the length scale estimates unless a prior gives additional shape to the surface.

MLE exhibits a higher variance and sensitivity to noise, particularly when fitting a model in high-dimensional spaces with scarce data. MAP, on the other hand, has a lower variance in the length-scale estimates but comes at the cost of bias unless accurate prior information is available. The MLL is given by

$$
\log p(\boldsymbol{y}|X,\boldsymbol{\theta}) = \underbrace{-\frac{1}{2}\boldsymbol{y}^{\mathsf{T}}\left(K(X,X)+\sigma_n^2 I\right)^{-1}\boldsymbol{y}}_{\text{data fit}}
$$
$$
\underbrace{-\frac{1}{2}\log|K(X,X)+\sigma_n^2 I|}_{\text{complexity penalty}} -\frac{n}{2}\log 2\pi \tag{3}
$$

The first and second terms are often called data fit and complexity penalty (Williams & Rasmussen, 2006). For more details, see Appendix A. This section explores the bias-variance trade-off between these two popular approaches for GP model fitting.

**MLE.** Fig. 6 shows the length scale obtained when using (blue) or MAP (orange) to fit a GP surrogate model with an $5/2$-ARD-Matérn to a realization drawn from an isotropic GP prior with length scale $\ell = 1$ and noise term $10^{-8}$. We examine a 10 and a 50-dimensional GP and repeat each experiment 50-times. As before, the gradient-based optimizer starts with an initial length scale of $\frac{\sqrt{d}}{10}$. We observe that the length scales estimated by MLE vary significantly less for the 10-dimensional function than for the 50-dimensional one. As we increase the number of observations on which the GP surrogate is conditioned, the variance of the estimated length scales decreases.

The dotted curves in the bottom row of Fig. 7 show the likelihood surface for MLE as specified in Eq. (3). The penalty term $\frac{1}{2}\log|K|$ and the data fit term

$-\frac{1}{2}\boldsymbol{y}^{\mathsf{T}}\left(K(X,X)+\sigma_n^2 I\right)^{-1}\boldsymbol{y}$ are shown in the first two rows, respectively; the constant term $-\frac{n}{2}\log 2\pi$ is omitted from the figure. We account for the different number of samples between the left and right figures by scaling the penalty, data fit, MLE, and MAP terms with $s^{-1}$. We show the surface for $s = 5$ (left) and $s = 50$ (right) data points. As the length scales increase, the entries of the kernel matrix increase. The determinant $|K|$ decays more quickly, and the penalty term decreases, adding a more distinct global trend to the likelihood surface. In the limit, $\ell \to \infty$, the kernel matrix becomes a matrix of ones, and the determinant becomes zero. In low dimensions, the data fit term decreases for long length scales, but the decreasing penalty compensates for this, resulting in a relatively flat MLL surface. The fast decay of the data fit term increases the "signal-to-noise" ratio, making it easier for the optimizer to converge in 10 than 100 dimensions. This can be seen by comparing the green and brown MLL curves in Fig. 7 for $s = 50$ samples. For more observations, MLL becomes smoother in all dimensions, as indicated by the left vs. right panel in Fig. 7.

**MAP.** MAP allows for incorporating prior beliefs about reasonable values for hyperparameters. However, practitioners often do not possess such prior information and hence resort to hyperpriors that reportedly perform well in benchmarks. Karvonen & Oates (2023) criticized this as an 'arbitrary' determination of hyperparameters.

The orange distributions in Fig. 6 show the average length scales obtained by MAP with a Gamma$(3, 6)$ prior, which has been the default in BoTorch before version 12.0. We use this prior as it has a substantial mass around its mode $1/3$,

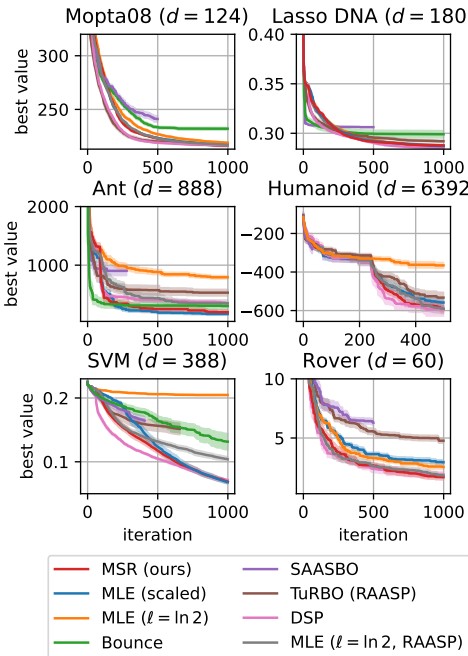

*Figure 8.* BO with the 'scaled' initialization of MLE performs comparably to the state-of-the-art in HDBO.

*Table 1.* Comparison of the simple BO methods used for the empirical evaluation.

| Method | length scale scaling | RAASP sampling |
|---|---|---|
| MSR | ✓(initial value) | ✓ |
| MLE (scaled) | ✓(initial value) | ✗ |
| MLE ($\ell = \ln 2$) | ✗ | ✗ |
| DSP | ✓(prior) | ✓ |

and we refer to this new BO method as 'MLE scaled'. The second method in the evaluation is the same BO method, but now using the default value $\ell = \ln 2$ of GPyTorch as initial length scale in MLE. This value is not scaled with $d$. Based on our analysis of the impact of random axis-aligned subspace perturbation (RAASP) sampling when optimizing the acquisition function (AF), we combine 'MLE scaled' with RAASP sampling and call the method 'MLE Scaled with RAASP' (MSR). We detail the RAASP sampling and the AF maximization in Appendix B. The next method, DSP, is the new default in BoTorch (Balandat et al., 2020), which uses a maximum a-posteriori estimation (MAP) estimate of length scales and initializes the optimization with the mode of the dimensionality-scaled length scale prior (DSP) (Hvarfner et al., 2024). Table 1 summarizes the methods with basic BO setups we use for the empirical evaluation.

We also compare against SAASBO (Eriksson & Jankowiak, 2021), TuRBO (Eriksson et al., 2019), and Bounce (Papenmeier et al., 2023). SAASBO has a large computational runtime. Hence, we run it only for 500 iterations and terminate runs that exceed 72 hours. That is why SAASBO has fewer evaluations for Ant and Humanoid.

Our benchmarks are the 124-dimensional soft-constrained version of the Mopta08 benchmark (Jones, 2008) introduced by Eriksson & Jankowiak (2021), the 180-dimensional Lasso-DNA (Šehić et al., 2022), the 388-dimensional SVM (Eriksson & Jankowiak, 2021), the 60-dimensional Rover (Eriksson et al., 2019), and two 888- and 6392-dimensional Mujoco benchmarks used by Hvarfner et al. (2024). The first four benchmarks are noise-free, while the others exhibit observational noise.

**MLE Works Well for HDBO.** Fig. 8 shows the performance of the BO methods on the four real-world applications. Each plot gives the average objective value of the best solution found so far in terms of the number of iterations. We show the ranking of the methods according to the final performance in Table 2 in Appendix C.1. The confidence bands indicate the standard error of the mean. MSR achieves competitive performance across all benchmarks, matching the SOTA DSP. Notably, MSR outperforms DSP on $124d$ Mopta08 and $888d$ Ant, and performs

simplifying our analysis compared to wider priors, which reduce the difference between MLE and MAP. Compared to MLE, the MAP estimates vary less but exhibit significant bias. This is pronounced for the 50-dimensional GP sample, where the MAP estimates for the length scales revert to the prior mode for 100, 200, 500, and 1000 initial samples. The solid lines in the lower row of Fig. 7 show the surface for the MAP estimation, using the same GP sample as for MLE. The log prior term adds additional curvature, resulting in length scale estimates of lower variance. This is particularly noticeable for little data (left column of Fig. 7) and consistent with Fig. 6. With more data, MLE and MAP become increasingly more similar, with MAP's log posterior decreasing faster for longer length scales due to the Gamma prior.

## 4. Discussion

**Experimental Setup and Benchmarks.** We propose a simple initialization for the gradient-based optimizer used for fitting the length scales of the Gaussian process (GP) surrogate *via* MLE and evaluate its performance for BO tasks. In what follows, we suppose a BO algorithm with a $^5/_2$-ARD-Matérn kernel and LogEI (Ament et al., 2024). To address the issue of vanishing gradients at the start of the MLE optimization, we choose the initial length scale as 0.1 and scale with $\sqrt{d}$ to account for the increasing distances of the randomly sampled design of experiments (DOE) points. Thus, the initial length scale used in the optimization is $\frac{\sqrt{d}}{10}$,

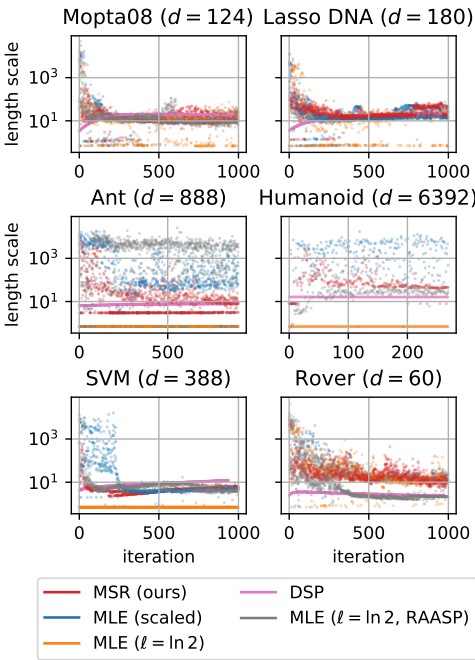

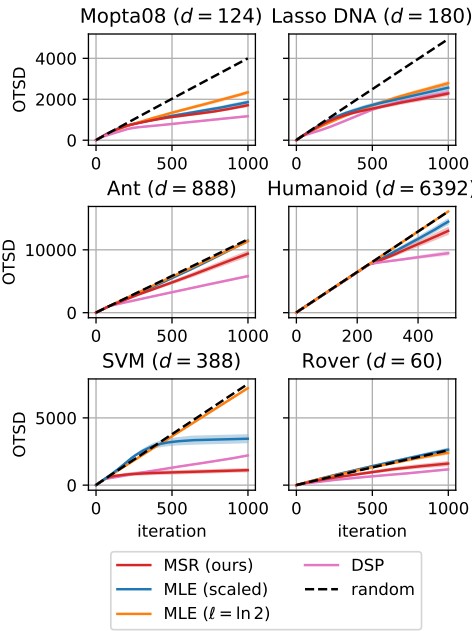

*Figure 9.* Average length scales of `MSR` and the other methods. RAASP sampling gives more consistent length scale estimates.

*Figure 10.* `DSP` exhibits the least exploration. MLE with fixed initial length scales performs like random search on `Ant` and `Humanoid`.

slightly worse than `DSP` other benchmarks. `Bounce` (Papenmeier et al., 2023) is consistently outperformed by MSR, MLE, and DSP but surpasses `SAASBO` (Eriksson & Jankowiak, 2021), especially on `Ant`. Although the constant length scale initialization ($\ell = \ln 2$) without RAASP achieves satisfactory results on lower-dimensional benchmarks such as $124d$ `Mopta08` and $180d$ `Lasso-DNA`, it fails on higher-dimensional benchmarks like $888d$ `Ant` and $6392d$ `Humanoid`. We attribute this breakdown to vanishing gradients as shown in Fig. 13 in Appendix C.2.

Fig. 9 compares the mean length scales per BO iteration, averaged over dimensions and 15 repetitions. For the 124-dimensional `Mopta08` and 180-dimensional `Lasso-DNA` applications, 'MLE ($\ell = \ln 2$)' learns length scales similar to the other methods. However, this constant initialization strategy fails to make progress from the initial value for the more high-dimensional problems in the bottom row of Fig. 9. We attribute this to vanishing gradients of the marginal log-likelihood (MLL) as discussed in Sec. 3 and highlighted in Fig. 13 in Appendix C.2.

**RAASP Reduces Variance.** Fig. 9 shows a surprising behavior for `DSP`. As one would anticipate, the estimated length scales are typically close to the mode of the hyperprior at the start of the optimization. However, they then converge to an even higher value on all benchmarks but the 6392-dimensional `Humanoid` benchmark. Furthermore, the deviation from the prior mode is more pronounced for

the lower-dimensional benchmarks, being in line with our analysis of the bias-variance trade-off in Sec. 3.3. At the beginning of its execution, the BO algorithm that uses MLE with scaled initial length scales ('MLE (scaled)') uses longer length scales than all other methods. The resulting estimates vary significantly for the high-dimensional `Ant` and `Humanoid` problems, supporting our analysis in Sec. 3.3 where we study the comparatively high variance of MLE compared to MAP.

The length scales obtained by `MSR` lie between the values of `DSP` and of 'MLE without RAASP sampling' (green dots, 'MLE (scaled)') for most benchmarks. An exception is `Ant` where `MSR` sometimes results in shorter length scales than `DSP`. Overall, the RAASP sampling, which is the only difference between `MSR` and 'MLE (scaled)', obtains more consistent length scale estimates.

**RAASP Promotes Locality.** Fig. 10 compares the amount of exploration that the algorithms perform through the lens of the observation traveling salesperson distance (OTSD) metric; see Section 3.2 for details on OTSD. We observe that `DSP` (blue curves) is the most exploitative method on all benchmarks, being in line with the fact that, after MLE with constant length scale initialization ('MLE ($\ell = \ln 2$)'), `DSP` has the shortest length scales on most benchmarks. The fact that 'MLE ($\ell = \ln 2$)' is the most *explorative* method, coinciding with the 'random search' line in Fig. 10, despite having the *shortest* length scales is explained by our

analysis in Sec. 3.2: the random initial points for the AF optimization are not further optimized if the gradient of the AF vanishes, because the method does not employ RAASP sampling. We observe that 'MLE ($\ell = \ln 2$)' does not learn longer length scales during the optimization for `Ant` and `Humanoid`, as indicated by the horizontal brown line in in Fig. 9. Thus, it does not recover later. For `Mopta08` and `Lasso-DNA`, which have a lower input dimension, the effect is less pronounced because 'MLE ($\ell = \ln 2$)' sometimes learns longer length scales that avoid vanishing gradients of the AF. MLE with scaled initial length scale values (green curves in Fig. 10) is the second-most explorative method. However, this is not due to vanishing gradients but caused by overly long length scales, shown as green dots in Fig. 9. `MSR` (red curves) is more explorative than `DSP` and more exploitative than 'MLE (scaled)', which does not use RAASP sampling. This is consistent with the shorter length scales of `MSR` (red dots in Fig. 9), confirming that `MSR` not only yields more consistent length scale estimates but also acts more local than its RAASP-sampling-free equivalent.

**Notes on Popular HDBO Benchmarks.** In Fig. 9, all methods converge to similarly long length scales on the `Mopta08` and `Lasso-DNA` benchmarks. This is likely attributable to a specific property of these popular benchmarks, as we discuss in Appendix D. In short, these benchmarks have a simple structure that benefits models with long length scales, posing the risk of algorithms being 'overfitted' to these benchmarks.

## 5. Conclusions and Future Work

Our analysis reveals underlying challenges in high-dimensional Bayesian optimization (HDBO) while offering practical insights for improving HDBO methods. We demonstrate that common approaches for fitting Gaussian processes (GPs) cause vanishing gradients in high dimensions. We propose an initialization for maximum likelihood estimation (MLE) that achieves state-of-the-art performance on established HDBO benchmarks without requiring the assumptions of maximum a-posteriori estimation. Finally, we provide empirical evidence that combining MLE with random axis-aligned subspace perturbation (RAASP) sampling reduces the variance of the length scale estimates and yields values closer to the ones learned by `DSP`, providing a fresh argument for the inclusion of RAASP sampling in HDBO.

In future work, we will continue to carefully vet popular benchmarks and propose novel, challenging benchmarks that preserve the traits of real-world applications. Furthermore, this work emphasizes the importance of numerical stability for the performance of Bayesian optimization (BO) in high dimensions. Thus, we propose approaching the development of models and acquisition functions (AFs) from this perspective.

Our work focuses on GP surrogate models but we will explore in how far our findings can be extended to other surrogate models, such as random forests or Bayesian neural networks.

Finally, we will explore how our findings help improve the performance of established techniques for HDBO by combining `MSR` with trust regions (TRs) (Eriksson et al., 2019), adaptive subspace embeddings (Papenmeier et al., 2022), or additive structures (Duvenaud et al., 2011).

## Impact Statement

This paper presents work that aims to advance the field of Machine Learning. There are many potential societal consequences of our work, none of which we feel must be specifically highlighted here.

## Acknowledgements

This project was partly supported by the Wallenberg AI, Autonomous Systems, and Software program (WASP) funded by the Knut and Alice Wallenberg Foundation. The computations were enabled by resources provided by the National Academic Infrastructure for Supercomputing in Sweden (NAISS), partially funded by the Swedish Research Council through grant agreement no. 2022-06725

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

# A. A Review of Gaussian Processes and Bayesian Optimization

## A.1. Gaussian Processes

Gaussian processes (GPs) model a distribution over functions, i.e., assume that $f$ is drawn from a GP: $f \sim \mathcal{GP}(m(\boldsymbol{x}), k(\boldsymbol{x}, \boldsymbol{x}') + \sigma_n^2 \mathbb{1}_{[\boldsymbol{x}=\boldsymbol{x}']})$ where $m$ and $k$ are the mean and covariance function of the GP, respectively (Williams & Rasmussen, 2006). Common kernel functions include the radial basis function (RBF) kernel:

$$k_{\text{RBF}}(\boldsymbol{x}, \boldsymbol{x}') = \sigma_f^2 \exp\left(-\frac{r}{2}\right), \tag{4}$$

or the $^5/_2$ automated relevance determination (ARD)-Matérn kernel:

$$k_{\text{Mat}^5/_2}(\boldsymbol{x}, \boldsymbol{x}') = \sigma_f^2 \left(1 + \sqrt{5r} + \frac{5r}{3}\right) \exp\left(-\sqrt{5r}\right) \tag{5}$$

with $r = \sum_{i=1}^d \frac{(x_i - x_i')^2}{\ell_i^2}$. Here, $\boldsymbol{\ell}$ is a $d$-dimensional vector of component-wise length scales. Thus, the kernel's number of hyperparameters (HPs) in Eq. (5) is $d+1$.

Given some training data $\mathcal{D} := \{(\boldsymbol{x}_1, y_1), \ldots, (\boldsymbol{x}_N, y_N)\}$, $X := (\boldsymbol{x}_1^\mathsf{T}, \ldots, \boldsymbol{x}_N^\mathsf{T})^\mathsf{T}$, $\boldsymbol{y} = (y_1 \ldots, y_N)^\mathsf{T}$, the function values $\boldsymbol{y}_*$ of a set of query points $X_*$ is normally distributed as

$$\boldsymbol{y}_* | X, \boldsymbol{y}, X_* \sim \mathcal{N}(K(X_*, X)(K(X, X) + \sigma_n^2 I)^{-1} \boldsymbol{y}, \tag{6}$$

$$K(X_*, X_*) - K(X_*, X)(K(X, X)^{-1} + \sigma_n^2 I)K(X, X_*)) \tag{7}$$

Let $\boldsymbol{\theta} = \{\sigma_n^2, \sigma_f^2, \boldsymbol{\ell}\}$ be the set of GP hyperparameters. The GP surrogate is then typically fitted by maximizing the marginal log-likelihood w.r.t. $\boldsymbol{\theta}$, also known as maximum likelihood estimation (MLE), i.e.

$$\boldsymbol{\theta}^* \in \arg\max_{\boldsymbol{\theta}} \log p(\boldsymbol{y} | X, \boldsymbol{\theta}) \tag{8}$$

$$\log p(\boldsymbol{y} | X, \boldsymbol{\theta}) = -\frac{1}{2} \boldsymbol{y}^\mathsf{T} \left(K(X, X) + \sigma_n^2 I\right)^{-1} \boldsymbol{y} - \frac{1}{2} \log|K(X, X) + \sigma_n^2 I| - \frac{n}{2} \log 2\pi \tag{9}$$

With a gradient-based approach, this is done by maximizing Eq. (9), which is usually multi-modal and difficult to optimize. The gradient of the marginal log-likelihood w.r.t. $\theta_i$ is given by

$$\frac{\partial}{\partial \theta_i} \log p(\boldsymbol{y} | X, \boldsymbol{\theta}) = \frac{1}{2} \boldsymbol{y}^\mathsf{T} \left(K + \sigma_n^2 I\right)^{-1} \frac{\partial K}{\partial \theta_i} \left(K + \sigma_n^2 I\right)^{-1} \boldsymbol{y} - \frac{1}{2} \text{tr}\left(\left(K + \sigma_n^2 I\right)^{-1} \frac{\partial K}{\partial \theta_i}\right), \tag{10}$$

where $\frac{\partial K}{\partial \theta_i}$ is the symmetric Jacobian matrix of partial derivatives w.r.t. $\theta_i$.

One often endows the GP hyperparameters with hyperpriors and seeks the mode of the posterior distribution, known as maximum a-posteriori estimation (MAP):

$$\boldsymbol{\theta}^* \in \arg\max_{\boldsymbol{\theta}} \log p(\boldsymbol{y} | X, \boldsymbol{\theta}) + \log p(\boldsymbol{\theta}). \tag{11}$$

## A.2. Bayesian Optimization

Bayesian optimization (BO) is an iterative approach, alternating between fitting the model and choosing query points. Query points are found by maximizing an acquisition function (AF), e.g., expected improvement (EI) (Mockus, 2005). The EI AF measures how much observing a point $\boldsymbol{x}$ is expected to improve over the best function value observed thus far. It is defined as

$$\text{EI}(\boldsymbol{x}) = \mathbb{E}_{f(\boldsymbol{x})}\left[[f(\boldsymbol{x}) - y^\star]_+\right] = (\mu_N(\boldsymbol{x}) - y^\star)\Phi(Z) + \sigma_N(\boldsymbol{x})\phi(Z) \tag{12}$$

with $Z = \frac{\mu(\boldsymbol{x}) - y^\star}{\sigma(\boldsymbol{x})}$, $\Phi$ and $\phi$ being the standard normal cumulative distribution function (CDF) and probability density function (PDF), and $\mu_N$ and $\sigma_N^2$ being the posterior mean and posterior variance at $\boldsymbol{x}$, i.e.,

$$\mu_N(\boldsymbol{x}) = K(\boldsymbol{x}, X)(K(X, X) + \sigma_n^2 I)^{-1} \boldsymbol{y} \tag{13}$$

$$\sigma_N^2(\boldsymbol{x}) = (k(\boldsymbol{x}, \boldsymbol{x}) + \sigma_n^2)(K(X, X) + \sigma_n^2 I)^{-1} K(X, \boldsymbol{x}) \tag{14}$$

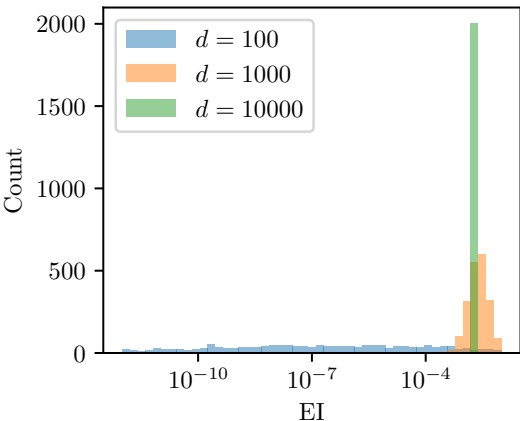

*Figure 11.* Distribution of EI values for GPs in various dimensionalities. When conditioning on the same amount of data points and maintaining the length scale as the dimensionality grows, the distribution of EI values becomes more peaked.

As discussed by (Ament et al., 2024), EI often suffers from vanishing gradients, which only worsens in high-dimensional spaces due to the plethora of flat regions. This is shown in Fig. 11. Here, we condition a GP on 100 random points in $[0, 1]^d$ for which we obtain function values by drawing from a GP prior with an isotropic RBF kernel with $\ell = 10$. We evaluate the AF on 2000 points drawn from a scrambled Sobol sequence and plot the histograms for various dimensionalities. As the dimensionality grows, there are more equal EI values, indicating flat regions in the AF and possible problems with vanishing gradients of the EI function. (Ament et al., 2024) propose `LogEI`, a numerically more stable version of EI that solves many of the numerical issues with EI.

## B. Additional Implementation Details

### B.1. Implementation of RAASP

We perturb the top 5% observations using a normal distribution with $\sigma = 10^{-3}$, truncated within $\mathcal{X}$. For $d \geq 20$, we also generate samples with only a subset of dimensions perturbed, each with a $\frac{20}{d}$ probability. Of $4m$ random samples, $2m$ are global samples from a scrambled Sobol sequence, $m$ are local samples around the top 5%, perturbing all dimensions, and $m$ are local samples around the top 5%, perturbing 20 dimensions on average if $d \geq 20$, or all dimensions if $d < 20$. We call the $2m$ local samples the RAASP samples. The starting points for the gradient-based optimization of the AF are drawn from the $4m$ overall samples using Boltzmann sampling (Ament et al., 2024).

### B.2. Optimization of the Acquisition Function

We use the `LogEI` AF (Ament et al., 2024) for its numerical stability. It is maximized by evaluating it on 512 scrambled Sobol points, then selecting five starting points via Boltzmann sampling for gradient-based optimization using L-BFGS-B, with up to 2000 iterations. The budget of 2000 iterations is rarely exhausted, as the optimizer typically converges much earlier (see Fig. 12). This aligns with previous work claiming that EI attains maxima close to good observations (Ament et al., 2024; Hvarfner et al., 2024) and our observation that the starting points for the gradient-based AF maximizer predominantly originate from the RAASP samples.

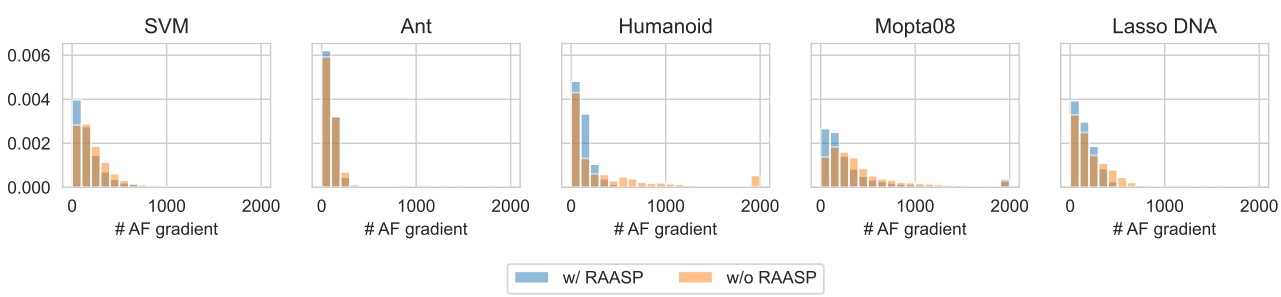

*Figure 12.* Number of gradient updates for the AF optimization for `MSR`, and with and without RAASP sampling. RAASP sampling reduces the number of gradient updates.

# C. Additional Experiments

## C.1. Ranking of Optimization Algorithms

|  | MSR | DSP | Bounce | MLE (scaled) | MLE ($\ell = \ln 2$) |
|---|---|---|---|---|---|
| `Mopta08` ($d = 124$) | 1 | 2 | 5 | 3 | 4 |
| `Lasso-DNA` ($d = 180$) | 4 | 1 | 5 | 3 | 2 |
| `Ant` ($d = 888$) | 2 | 4 | 3 | 1 | 5 |
| `Humanoid` ($d = 6392$) | 2 | 1 | - | 3 | 4 |

*Table 2.* Ranking for the different optimizers on the benchmark problems according to their final performance. `SAASBO` is excluded from the comparison as it was not run for the entirety of the optimization; `Bounce` ran into memory issues on `Humanoid` and, therefore, does not have a rank on this benchmark.

## C.2. MLE Gradients for Real-World Experiments

Complementing our analysis in Sec. 4, Fig. 13 shows the average absolute gradients of the different MLE methods, including our proposed `MSR` method. The constant length scale initialization ('MLE ($\ell = \ln 2$)') is the only method consistently exhibiting vanishing gradients on the 888-dimensional `Ant` and the 6392-dimensional `Humanoid` problems as depicted by the solid orange lines for those benchmarks. On `Mopta08` and `Lasso DNA`, all methods have non-vanishing gradients. Furthermore, both MLE methods scaling the initial length scale do not suffer from vanishing gradients on `Ant` and `Humanoid`.

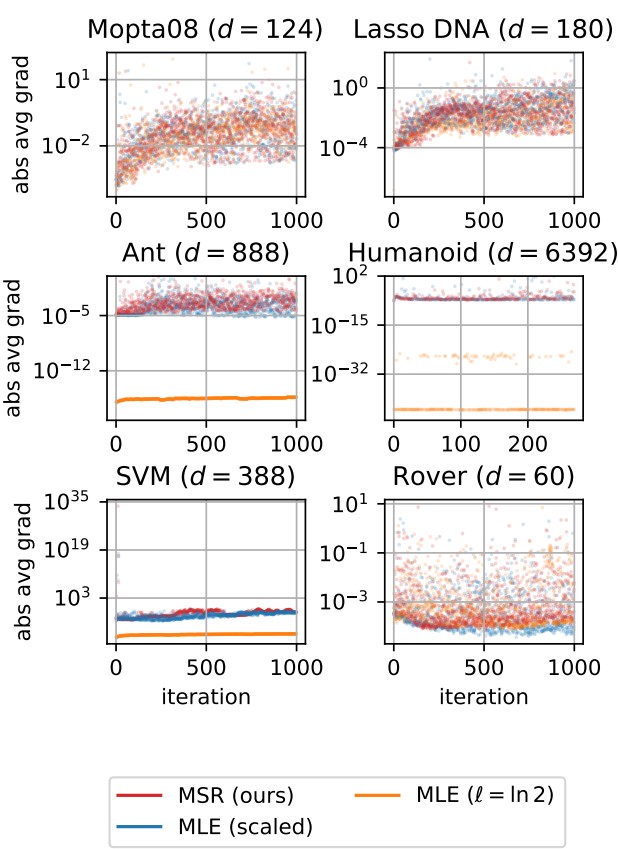

*Figure 13.* Mean absolute value of the gradients for the different MLE methods, including the proposed `MSR`. The constant length scale initialization exhibits vanishing gradients for the high-dimensional `Ant` and `Humanoid` problems.

## C.3. High-dimensional Benchmark Functions

By the no-free-lunch theorem (Wolpert & Macready, 1997), the relative performance of an optimization algorithm depends on the properties of the problems it operates on. Here, we show that for several benchmark problems, no state-of-the-art algorithm strictly dominates the other methods.

Table 3 shows the relative performances of `CMA-ES`, `DSP`, and `TuRBO` after 1000 optimization steps on the 100-dimensional versions of the `Levy`, `Schwefel`, and `Griewank` benchmarks. We evaluate `Levy` in the bounds $[-10, 10]$, `Schwefel` in the bounds $[-500, 500]$, and `Griewank` in the bounds $[-600, 600]$ by scaling from the unit hypercube in which the GP operates to the respective bounds before evaluating the function.

To better understand the reason for the performance differences, we study the observation traveling salesperson distance (OTSD) for the different functions, shown in Fig. 14. Plots of the 2-dimensional versions of all three benchmarks are shown in Fig. 17 and the performance and OTSD plot of 100-dimensional `Levy` figure can be found in Fig. 15. `CMA-ES` shows the lowest level of exploration and has the lowest OTSD on the `Schwefel` function, where it outperforms the two other algorithms. On `Griewank` (see Fig. 16), `DSP` has the highest average OTSD performs best while the less explorative `CMA-ES` shows the worst performance. For `Levy`, both `TuRBO` and `DSP` are relatively explorative and outperform `CMA-ES` by a considerable margin. We conclude that more explorative algorithms are advantageous on the benchmarks with a clear global trend like `Griewank`, which resembles a paraboloid, and `Levy`, which has a parabolic shape along the $x_1$ dimension (see Fig. 17). In contrast, the `Schwefel` benchmark is more "stationary" in that a point's function value depends less on that point's absolute position in the space. Noteworthy, stationarity as assumed by GP models with a stationary covariance function, which, by far, are the most common covariance functions for high-dimensional Bayesian optimization (HDBO). A more local approach such as `CMA-ES` is beneficial on this highly multi-modal benchmark.

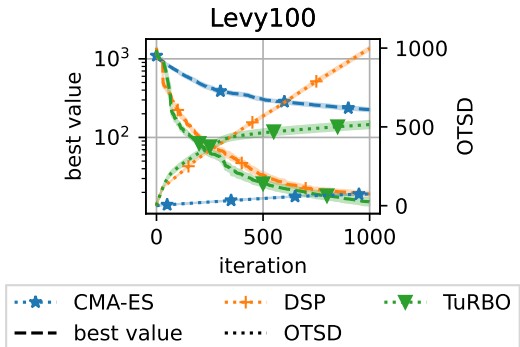

*Figure 15.* OTSD (solid lines) and performance curves (dashed lines) of the 100-dimensional `Levy` function

Figs. 15 and 16 show the OTSD and performance plots for the 100-dimensional `Levy` and `Griewank` functions. Fig. 17 shows the two-dimensional versions of the `Levy`, `Griewank`, and `Schwefel` benchmark functions.

| Benchmark | Rank 1 | Rank 2 | Rank 3 |
|---|---|---|---|
| Levy100 | TuRBO | DSP | CMA-ES |
| Schwefel100 | CMA-ES | DSP | TuRBO |
| Griewank100 | DSP | TuRBO | CMA-ES |

*Table 3.* Relative performances of `CMA-ES`, `DSP`, and `TuRBO` after optimizing for 1000 iterations averaged over 10 repetitions. No algorithm performs best for all benchmarks.

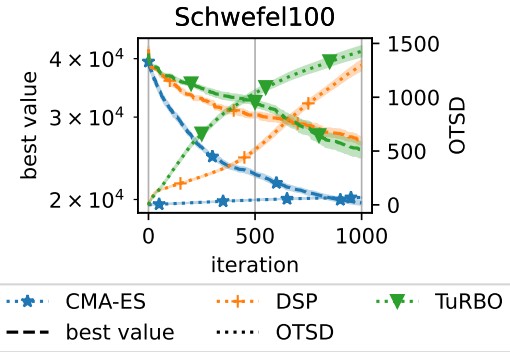

Figure 14. OTSD (solid lines) and performance curves (dashed lines) of the 100-dimensional `Schwefel` function

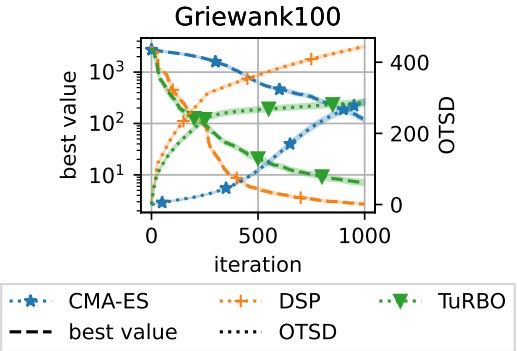

Figure 16. OTSD (solid lines) and performance curves (dashed lines) of the 100-dimensional `Griewank` function

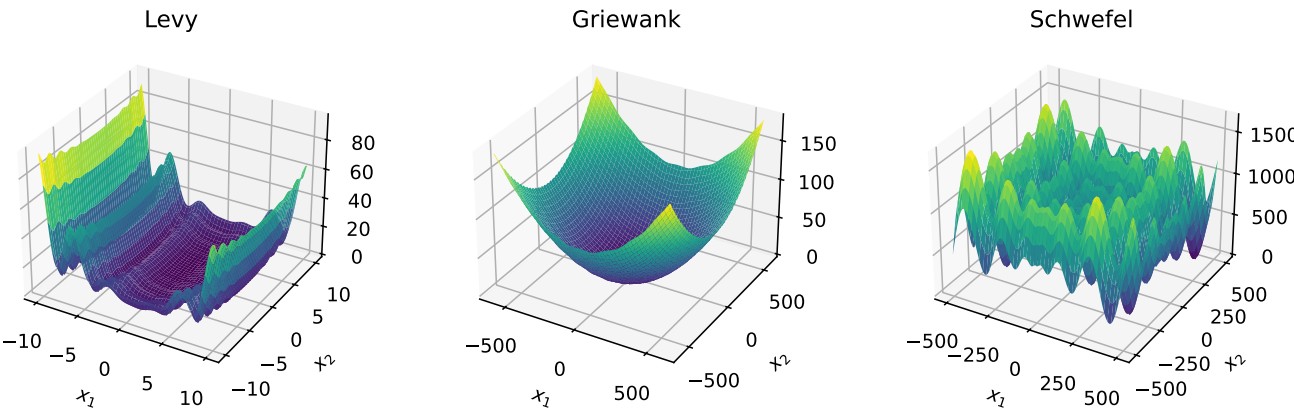

Figure 17. The two-dimensional versions of the `Levy`, `Griewank`, and `Schwefel` benchmark functions used above.

## C.4. Hard Optimization Problems

We reiterate that the curse of dimensionality (COD) remains a reality and exists even for low-dimensional problems. Fig. 18 shows 100 evaluations made by EI on a 2-dimensional GP prior sample as the benchmark function. There is no model mismatch; the length scales of the surrogate model are set to the correct value ($\ell = 0.025$). However, EI operates locally and fails to find the global optimum (marked by a red cross).

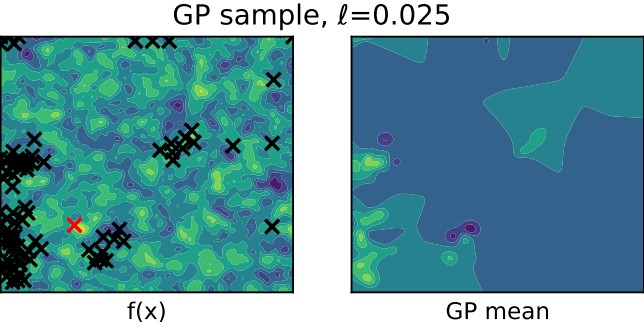

*Figure 18.* `LogEI` run on a two-dimensional GP prior sample for 100 evaluations. The right panel shows the posterior mean at the end of the optimization. For highly multimodal benchmarks, EI reverts to a local search behavior and does not obtain a global optimum (red cross).

## D. Popular Benchmarks Seem Simpler Than Expected

This section examines two popular high-dimensional benchmarks, the $180d$ Lasso-DNA and $124d$ Mopta08. In what follows, we will demonstrate that many input variables seem to have little influence on the objective value. These empirical findings suggest that these benchmarks are not truly as high-dimensional as their nominal number of input variables might suggest, potentially misleading the perceived difficulty of these benchmarks and confounding the assessment of what state-of-the-art algorithms will be able to accomplish in practice.

We begin by collecting the top 10% best points identified by the SOTA algorithm using dimensionality-scaled length scale prior (DSP) (Hvarfner et al., 2024) for each benchmark. For each dimension $x_i$ separately, we then count how often these points lie on the boundary of the search space. If over half of the points place $x_i$ on the boundary, we label $x_i$ as *secondary*, and as *dominant* otherwise. To ensure consistency across runs, we perform 15 repetitions and consider those dimensions that have been identified as dominant in eight or more of the repetitions.

Table 4 reports the number of dominant and secondary dimensions. In both benchmarks, a large fraction of the dimensions are classified as *secondary*, meaning that the best solutions obtained by DSP method set the corresponding input variables to value near the boundary of the search space. Note that our finding is directionally aligned with the 43 active dimensions that Šehić et al. (2022) reported for Lasso-DNA, using a different method to estimate the number of relevant dimensions.

We further investigate the impact of each set of dimensions by replacing, at each iteration, either the dominant dimensions ($f_{\text{dominant}}$) or the secondary dimensions ($f_{\text{secondary}}$) with randomly chosen binary values. The rows $\bar{f}_{\text{dominant}}$ and $\bar{f}_{\text{secondary}}$ in Table 4 indicate the corresponding average objective values (with standard errors in parentheses). Replacing secondary dimensions impairs the result only slightly, whereas randomizing the dominant dimensions yields a markedly larger performance drop. A two-sided Wilcoxon test shows that all differences are statistically significant at $p < 0.001$.

Next, we examine how Gaussian process (GP) models account for this structure during HDBO. As illustrated in Fig. 19, the estimated length scales of secondary dimensions are typically large, which is consistent with the observation that they influence the (predicted) objective value less. Intuitively, if a dimension's optimal value often lies at the boundary, the GP can assign it a very large length scale, learning the trend toward the boundary with few evaluations and leaving more of the budget to explore the truly dominant dimensions. In effect, the BO loop focuses on those dimensions that genuinely drive performance.

Below, we demonstrate that for these problems, BO will set most of the input variables to exactly zero or one –that is, to the boundary of the search space– regardless of whether the GP is fit by MLE or MAP. Thus, we conclude that the task effectively has a lower dimensionality than its nominal number of input variables. While this property was already recognized for Lasso-DNA, our results demonstrate for the first time that it also applies to Mopta08. Furthermore, it is interesting to note that a simple GP surrogate fitted via MLE or MAP can leverage this structure, a behavior that had not been previously observed.

Above, we showed that 1) dimensions of the best points observed by MLE and MAP that predominantly lie on the border have significantly longer length scales than the "dominant" dimensions, and 2) most of the dimensions have marginal impact. MLE shows a similar behavior and is omitted for brevity. We further omit the 388-dimensional SVM benchmark as it is known to have a low-dimensional effective subspace (Eriksson & Jankowiak, 2021).

To complement our analysis, we show that MLE and MAP assign dominant dimensions mainly values at the boundary. This

| | Lasso-DNA (d=180) | Mopta08 (d=124) |
|---|---|---|
| # dominant | 69.33 | 30.80 |
| # secondary | 110.67 | 93.20 |
| $\bar{f}_{\text{dominant}}$ | $0.315\,(\pm 4 \times 10^{-4})$ | $328.824\,(\pm 1.920)$ |
| $\bar{f}_{\text{secondary}}$ | $0.445\,(\pm 4 \times 10^{-3})$ | $430.474\,(\pm 8.164)$ |
| $\bar{f}_{\text{rand}}$ | $0.410\,(\pm 3.3 \times 10^{-2})$ | $403.428\,(\pm 48.976)$ |

*Table 4.* The number of dimensions identified as dominant and secondary and the average function values obtained when replacing the dominant ($\bar{f}_1$) or secondary ($\bar{f}_2$) parameters with uniformly random values. The average values and standard deviations for uniformly random points are shown as $\bar{f}_{\text{rand}}$. Replacing secondary parameters harms performance considerably less than replacing dominant parameters.

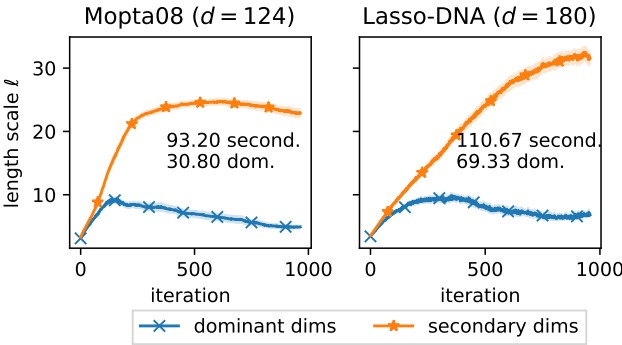

*Figure 19.* The mean average length scales of "dominant" and "secondary" dimensions for the `Mopta08` (left) and `Lasso-DNA` (right) benchmarks for DSP.

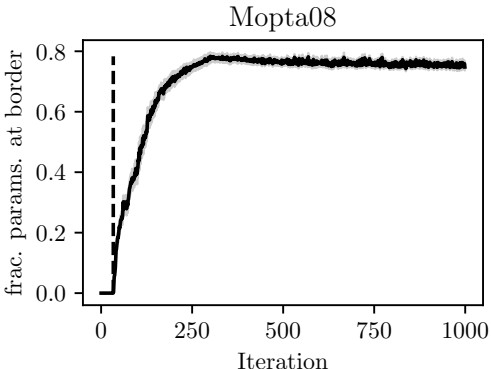
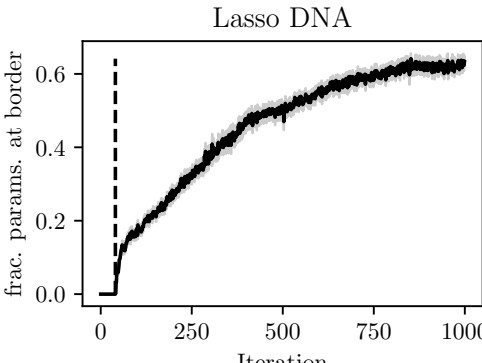

*Figure 20.* Fraction of dimensions set to a value at the border (0 or 1) by DSP. The shaded area shows the standard error of the mean across 15 repetitions.

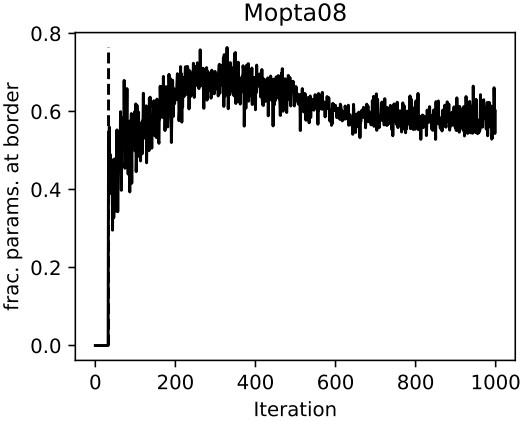
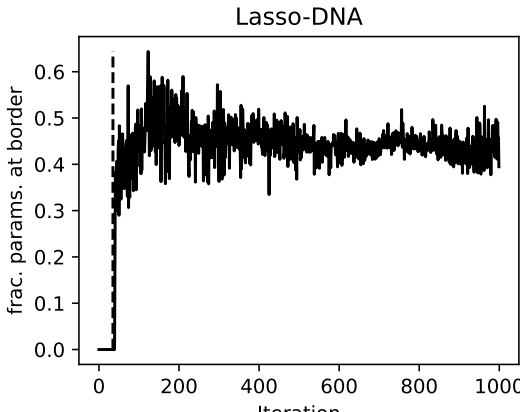

*Figure 21.* Fraction of dimensions set to a value at the border (0 or 1) by our MLE method. The shaded area shows the standard error of the mean across 15 repetitions.

indicates that the GP model actually makes use of the specific characteristics of these benchmarks.

Figs. 20 and 21 further show that BO consistently evaluates a large share of the parameters at the border. Fig. 20 shows this for DSP by (Hvarfner et al., 2024) whereas Fig. 21 uses MLE as proposed in Section 4. The general trend is that, during the course of the optimization, increasingly many parameters are evaluated at the border, which is consistent with Fig. 19.

We thus argue that two HDBO benchmarks do not fully capture the complexity of HDBO because of this simple structure. While this is to be expected for `Lasso-DNA` and `SVM`, this property has not been discussed for the soft-constrained `Mopta08` benchmark introduced in (Eriksson et al., 2019) to the best of our knowledge.

