# OpenReview forum: "Understanding High-Dimensional Bayesian Optimization"
_ICML.cc/2025/Conference — ICML 2025 poster_

### Official Review · Reviewer_bU1B · 2025-03-10

**Overall Recommendation:** 4

**Summary:**

This paper focuses on High-Dimensional Bayesian Optimization (HDBO). More precisely, it seeks to explain the good performance of vanilla BO for some high-dimensional tasks that has been observed by recent works. To do so, the paper empirically studies the vanishing gradients that appear when fitting a high-dimensional GP or when maximizing an acquisition function and derives multiple insights about the behavior of BO methods in high-dimensional input spaces.

The paper also proposes ways to mitigate the negative effects of vanishing gradients, such as Random Axis-Aligned Subspace Perturbations (RAASP) sampling or scaled Maximum Likelihood Estimation (MLE), and illustrates their benefits on empirical experiments.

**Claims And Evidence:**

This paper derives intriguing insights about HDBO and an interesting explanation about why recent HDBO methods perform well in high-dimensional settings.

However, I believe that "understanding HDBO" or "identifying fundamental challenges" are very strong and general claims, however this paper relies exclusively on empirical observations to support them. The results clearly point in an interesting direction, but more theoretical research is needed for, e.g., derive an upper bound on the magnitude of the gradients for the MLE and the acquisition function regardless of the objective function or quantify the impact of RAASP sampling on the magnitude of the gradients / on the regret of the algorithm.

**Essential References Not Discussed:**

I would have liked to see more recent papers on the additivity assumptions (e.g., [1, 2]), especially [2] which is the most recent and general work on the subject.

However, these are only useful in the contextualization of the paper. As far as I know, the paper properly discusses prior work.

**References**

[1] Hoang, T. N., Hoang, Q. M., Ouyang, R., & Low, K. H. (2018, April). Decentralized high-dimensional Bayesian optimization with factor graphs. In Proceedings of the AAAI Conference on Artificial Intelligence (Vol. 32, No. 1).

[2] Bardou, A., Thiran, P., & Begin, T. (2024). Relaxing the Additivity Constraints in Decentralized No-Regret High-Dimensional Bayesian Optimization. In The Twelfth International Conference on Learning Representations.

**Experimental Designs Or Analyses:**

The paper provides a lot of empirical observations and visualizations to support their conclusions on HDBO. All of them are insightful and sound.

**Methods And Evaluation Criteria:**

The paper identifies vanishing gradients as the main reason why vanilla BO does not perform well in high-dimensional input spaces and proposes a method that seems to mitigate this issue.

The method is compared against common strategies to learn the model hyperparameters and optimize high-dimensional objective functions, on high-dimensional problems and using multiple evaluation criteria.

Overall, this makes a lot of sense for the problem at hand.

**Other Comments Or Suggestions:**

As mentioned before, I would suggest to rephrase some claims to point out the empirical / exploratory nature of the research conducted in this paper.

Additionnally, here is a list of typos:
* in the abstract, "investigates the 'why'.'",
* "acquisition function" and its acronym "AF" are used interchangeably throughout the paper,
* on page 5, "more spread more evenly",
* Undefined reference to section on page 16.

**Other Strengths And Weaknesses:**

I would like to point out that the paper is well-written, I actually enjoyed reading it.

I have no other major strength / weakness to report.

**Questions For Authors:**

Here is a question to spark the discussion with the authors:

1) Have you thought about the implications of vanishing gradients (and the use of mitigating techniques like RAASP sampling) on the asymptotic performance of HDBO algorithms? Do you think that it is still possible to derive a sublinear regret bound under RAASP?

**Relation To Broader Scientific Literature:**

As I mentioned before, the paper provides very relevant insights on HDBO. I think this exploratory research is valuable to the HDBO community and should spark theoretical research efforts to better study the phenomena highlighted in this paper.

**Theoretical Claims:**

Excluding the (overly) general claims made in the title and the abstract of the paper, this work does not make any theoretical claim.

---

> ### Author Rebuttal · Authors · 2025-03-31
>
> We appreciate the reviewer's positive feedback regarding the insights and empirical observations presented in our paper. We are glad to hear that our work is seen as valuable to the HDBO community, well-written, and enjoyable.
>
> **General claims in the title and abstract**
>
> We will tone down the "identifying fundamental challenges" part of the text and edit the abstract and introduction to communicate the empirical focus of our work to set the right expectations for the reader.
>
> **Sublinear regret bounds**
>
> Most theoretical analyses of Bayesian Optimization, including those that derive sublinear regret bounds, assume that the acquisition function can be maximized exactly [1-4]. RAASP, as a technique to avoid vanishing gradients of the acquisition function, is orthogonal to this. Arguably, it was a long-standing belief that maximizing the acquisition function is simple enough for proofs not to focus on this aspect [5]. Our analysis shows that the error introduced during the acquisition function maximization deserves more attention, but we leave the derivation of sublinear regret bounds under RAASP for future work.
>
> **References**
>
> We appreciate the suggestion to include more recent references on additivity assumptions. We will incorporate the recommended papers to contextualize our work better.
>
> **Typos and minor issues**
>
> We will correct the typos and clarify the interchangeable use of 'acquisition function' and 'AF.' We will also fix the undefined reference and ensure consistency throughout the paper.
>
>
> [1] Srinivas, Niranjan, et al. "Gaussian Process Optimization in the Bandit Setting: No Regret and Experimental Design." Proceedings of the 27th International Conference on Machine Learning.
>
> [2] Wu, Kaiwen, et al. "The behavior and convergence of local bayesian optimization." Advances in neural information processing systems 36 (2023)
>
> [3] Scarlett, Jonathan. "Tight regret bounds for Bayesian optimization in one dimension." International Conference on Machine Learning.
>
> [4] Wang, Ziyu, and Nando de Freitas. "Theoretical analysis of Bayesian optimisation with unknown Gaussian process hyper-parameters." arXiv preprint arXiv:1406.7758 (2014).
>
> [5] Bull, Adam D. "Convergence rates of efficient global optimization algorithms." Journal of Machine Learning Research 12.10 (2011).

---

### Official Review · Reviewer_MivE · 2025-03-11

**Overall Recommendation:** 3

**Summary:**

This paper considers High-dimensional Bayesian Optimization and thoroughly studies the use of vanilla Gaussian Processes (GP) for that matter. A recent line of work illustrated how simple tweaks can lead to considerable improvements in optimization outcomes, with GPs outperforming complex state-of-the-art baselines. Such tweaks rely on proper lengthscale prior scaling and clever initialization strategies for Marginal log likelihood optimization. The paper thoroughly analyzes the effect that these modifications have on lengthscale optimization, finding that they effectively induce different kinds of search behavior in high dimensions, e.g., local or random. Based on these findings, a simple variant is proposed, and various benchmarks assessing its performances are provided.

**Claims And Evidence:**

Yes.

**Essential References Not Discussed:**

Not that I am aware of.

**Experimental Designs Or Analyses:**

I do not see any particular issue with the experiments and their subsequent analysis.

**Methods And Evaluation Criteria:**

- The high-dimensional benchmarks are restricted to vanilla Gaussian Processes and do not involve other surrogates like Bayesian Neural Networks, nor baselines using complex nonlinear embeddings like VAEs. Given the scope of the paper, which focuses on explaining how GP lengthscales behave throughout optimization in high-dimensional spaces, I have no issues with that.

- The high-dimensional datasets employed for evaluation are standard. However, the authors highlight that the high-dimensional benchmarks employed by the community, which they also use in their evaluation, might not be "that high-dimensional", as detailed in Appendix C. They argue that "these benchmarks have a simplifying structure that benefits models with long length scales, posing the risk of algorithms being ‘overfitted’ to these benchmarks."

**Other Comments Or Suggestions:**

Fig 5, typo: The behavior of short-length-scale GPs
reverts to local search (ℓ = 0.05 in the left panel) with RAASP
sampling and to **local search** without RAASP sampling (ℓ = 0.05
and ℓ = 0.28 in the right panel).

**Other Strengths And Weaknesses:**

The paper is well-written and well-organized. The technical novelty is light, but that is not the goal here. Instead, the paper does a great job at dissecting the complex behavior of GP lengthscale optimization in high-dimensional BO. As a practitioner myself, I appreciate the insights delivered by this analysis, and I believe the community will highly benefit from them.

**Questions For Authors:**

I do not have any questions.

**Relation To Broader Scientific Literature:**

This paper is part of a recent trend, consisting of reverting back to simple methods for high-dimensional black-box optimization. The work it builds on is cited, and the field of high-dimensional BO as a whole is well-described in the related work.

**Theoretical Claims:**

There is no theoretical claim.

---

> ### Author Rebuttal · Authors · 2025-03-31
>
> We would like to thank the reviewer for their positive feedback, particularly for recognizing the thorough analysis of Gaussian Processes in high-dimensional Bayesian Optimization (HDBO), as well as the clarity and practical insights provided in the paper. We appreciate the reviewer's thoughtful comments and considered each point to enhance the clarity and impact of our manuscript, as detailed below.
>
> **Focus on Gaussian Processes**
>
> We will clarify in the discussion section that our analysis is specifically tailored to GPs and does not extend to other surrogates. This focus allows us to provide a deep dive into the behavior of GP length scales in high-dimensional spaces. However, studying other surrogates is an important avenue for future work. The vanishing gradients we observe are primarily related to popular stationary kernels for which the covariance of two points vanishes quickly with their distance. Other surrogates, such as random forests or Bayesian neural networks, use different techniques to relate input and output similarity and may show different susceptibility to vanishing gradients.

---

### Official Review · Reviewer_9KgT · 2025-03-13

**Overall Recommendation:** 4

**Summary:**

The paper investigates the perfomance of BO for high-dimensional problems, building upon a recent focus in the literature on working out how to make standard BO frameworks work better in high-dimensions, rather than proposing entirely new frameworks. Lots of empirical evidence is provided to justify the simple solution they propose, i.e. clever lengthscale initialisation + local acquisition optimisation. Along the way, lots of intersting/practically important observations are made.

## update after rebuttal

Thanks for the rebuttal. I hope to see all these extra details make it into the final version. I have raised my score.

**Claims And Evidence:**

The paper claims that the real issue (in contrast to that proposed in recent works) when fitting Gps in high-dimension is to do with vanishing gradients in the MLL. Significant evidence is provided to support this claim, including targeted analysis as well as full BO pipeline runs.

**Essential References Not Discussed:**

N/A

**Experimental Designs Or Analyses:**

I think a fair few key pieces of information are missing (or need better signposting) in order to claim reproducibility:
1. exact details for RASSP implementation (see below, i.e. Gaussian proposal params, and budget e.t.c)
2. details for the acquisition function used in the experiments
3. details on how the acquisition function is optimised when not doing RASSP

**Methods And Evaluation Criteria:**

For what is largely empirical work, the evaluations are somewhat limited with regards to ablations. For example, it would be powerful to see results for differnet acqusition function optimisation budgets (and indeed some actualy details on how this is perfomed....).

**Other Comments Or Suggestions:**

It would be nice for a full discusson of RASSP (i.e. what exact params it has for its Gaussian proposal e.t.c), rather than just referencing Botorch (apologies If i missed this).

I know the aim of the paper is to investigate how standard BO algorithms can be adapted to high-dim BO, but it would be a stronger sell if you included some results with more of the popular high-dim BO algorithms, like TURBO e.t.c

**Other Strengths And Weaknesses:**

The paper is very well-written and was a pleasure to read. I feel that I learned a couple of things along the journey, identifying many of the strange behaviours that I have observed empirically when applying BO in practice.

**Questions For Authors:**

Improved discussion/experimentation around acquisition optimisation budgets and how they interplay with the proposed method would encourage me to raise my score.

**Relation To Broader Scientific Literature:**

This work builds upon recent work selling the use of sutably modified traditional BO algorithms for high-dimensional BO. This strand of work has oppurtunities for high-impact, especially as the proposed algorithm is so simple/easy to apply in exisiting frameworks/pipelines.

I do worry a little bit about the utility of this work above the recent papers in this field (e.g. the Vanilla BO paper referenced throughout). I agree that the method proposed here seems to be better motivated + more generally applicable, but in absolute terms, the contribution is not huge.

**Theoretical Claims:**

N/A

---

> ### Author Rebuttal · Authors · 2025-03-31
>
> We appreciate the positive feedback regarding the clarity of our writing and the potential impact of our simple yet effective solution. We implemented enhancements to clarify our methodology and bolster the impact of our findings.
>
> **Implementation of RAASP**
>
> We will improve the description of the RAASP implementation in Section 3.2. We will further clarify that MSR uses BoTorch’s RAASP implementation. Since we cannot upload a revised manuscript, we will briefly outline the details here.
>
> We perturb the top 5% observations using a normal distribution with $\sigma=0.001$, truncated within $\mathcal{X}$. For $D\geq 20$, we also generate samples with only a subset of dimensions perturbed, each with a $\frac{20}{D}$ probability. Of $4m$ random samples, $2m$ are global samples from a scrambled Sobol sequence, $m$ are local samples around the top 5%, perturbing all dimensions, and $m$ are local samples around the top 5%, perturbing 20 dimensions on average if $D\geq 20$, or all dimensions if $D<20$. We call the $2m$ local samples the RAASP samples. The starting points for the gradient-based optimization of the acquisition function are drawn from the $4m$ overall samples using Boltzmann sampling [1].
>
> **Details on the AF and its optimization**
>
> We will expand the methodology section to detail the AF and optimization process without RAASP to improve reproducibility and clarity. We briefly outline the details below.
>
> We use the LogEI AF [1] for its numerical stability. It is maximized by evaluating it on 512 scrambled Sobol points, then selecting five starting points via Boltzmann sampling for gradient-based optimization using L-BFGS-B, with up to 2000 iterations.
>
> **Magnitude of the contribution**
>
> This work sheds new light on key challenges and fundamental issues in HDBO, like vanishing gradients in GP model fitting and AFs. These are critical for understanding the limitations of popular algorithms and were not part of prior work. This revision of prior concepts will prompt broader comfort for BO practitioners to use methods such as DSP and MSR.  We expect that our identification of a more straightforward, high-performing implementation - supported by uncovering fundamental principles and efficiencies - enables researchers to investigate novel algorithmic variations and applications. MSR introduces a simple initialization method for GP length scales using MLE that scales with the problem dimensionality, eliminating the need for prior beliefs (Section 4). We further identify the importance of RAASP for avoiding vanishing gradients and reducing variance in length scale estimates (Sections 3.2 and 4). We also show that the MLE estimates have high variance while MAP estimates are biased, relating it to the variance-bias tradeoff (Section 3.3).
>
> **Comparisons with other HDBO algorithms**
>
> We now compare our method with TuRBO (https://ibb.co/jkx3MXN1), which is competitive on Mopta08 and Lasso-DNA and performs well on Humanoid for the first half of the optimization but stagnates in the second half.
>
> **Discussion of AF optimization budgets**
>
> We interpret this comment as 1) gradient descent budget, 2) RAASP budget, and 3) multi-start initialization for the AF. We will clarify this in the manuscript by adding these explanations.
>
> 1. We capped the number of gradient descent steps to 2000, the BoTorch default budget. We will add to the manuscript that this budget is rarely depleted, as shown in this histogram plot https://ibb.co/bjQQvskj
> 2. When augmenting the candidate set of starting points for the AF maximization with RAASP samples, we see fewer gradient descent steps required to optimize the AF. This aligns with previous work claiming that EI attains maxima close to good observations [1,2] and our observation that the starting points for the gradient-based AF maximizer predominantly originate from the RAASP samples. We will add this explanation to the manuscript and pose the question of how much optimization is sufficient in the future work section.
> 3. Evaluating the AF on more initial points before selecting starting points can reduce the risk of vanishing gradients. This figure (https://ibb.co/XkL4yCpn) illustrates this by plotting the fraction of points that moved a positive distance after maximizing the AF against the number of initial points. We maximized LogEI on a 50-dimensional GP prior sample ($\ell=0.2$, isotropic kernel) and repeated the experiment 50 times. With $2^{11}$ random samples, only 2% of final candidates moved a non-zero distance. With $2^{17}$ samples, this increased to 50%. Using RAASP sampling with just 512 raw samples, none of the starting points exhibited vanishing gradients across all 50 repetitions.
>
> These discussions will corroborate our arguments, and we will add them to the manuscript.
>
> [1] Ament et al. "Unexpected improvements to expected improvement for bayesian optimization." NeurIPS 2023
>
> [2] Hvarfner et al. "Vanilla Bayesian Optimization Performs Great in High Dimensions." ICML 2024

---

### Official Review · Reviewer_xFof · 2025-03-14

**Overall Recommendation:** 3

**Summary:**

The paper provides an investigation on the problems of BO in high dimensions (the curse of dimensionality). First is the problem of vanishing gradients when modelling GP, which makes some GP hyperparameters, such as lengthscales cannot be updated. Based on recent work of Hvarfner et al., 2024, the authors find that increasing the lengthscale prior such that it scales with problem’s dimensionality can help mitigate the problem. The second mentioned problem is the vanishing gradients when optimizing AF. The author mentions 2 solutions: numerically stable LogEI acquisition function (Ament et al., 2024) and RAASP sampling strategy from TuRBO (Eriksson et al., 2019). Third, the authors provide some insights on the trade-off between bias and variance when fitting GP, particularly a comparison between MLE and MAP. While MLE has higher variance, MAP exhibits a bias towards the prior, which can be devastating in case of inaccurate prior assumption. Finally, the authors combine these findings into a proposed method MSR (MLE Scaled with RAASP) and compare them against several variants and other recent SOTAs.

**Claims And Evidence:**

All of the three claims are supported by clear experimental results and also in line with previous paper findings.

**Essential References Not Discussed:**

To my knowledge, all of the essential related works are discussed.

**Experimental Designs Or Analyses:**

The baseline SOTAs, including SAASBO, Bounce and DSP, are reasonable comparisons. However, I think it would be better if the author could include:

•	A baseline that also use RAASP sampling strategy. For example, TuRBO (Eriksson et al., 2019) or MSR variant that does not use scaled lengthscale. Note that Bounce (different from its previous version BAxUS, does not use RAASP anymore).

•	RDUCB (Ziomek & Ammar, 2023) are included, a recent SOTA leveraging additivity assumption of f. This is to confirm the performance against other HDBO approaches.

**Methods And Evaluation Criteria:**

•	There are only four benchmark problems used in the experimental results, while common BO papers have at least 6.

•	To make it worse, the authors criticize (in Section 4 – Notes on Common HDBO Benchmarks) that LassoDNA and Mopta08 problems have structures that pose the risk for HDBO algorithms but then fail to use other benchmarks to support the proposed methods.

**Other Comments Or Suggestions:**

Minor:

The writing in Section 4 – Experimental Setup and Benchmarks are hard to follow. The authors mention “the second method” and “the fifth method”, but do not mention the other methods.

**Other Strengths And Weaknesses:**

Strengths:

-	The discussed issues of HDBO are worth investigating and are systematically summarized.
-	The proposed methodology is simple yet effective, even when compared to recent more sophisticated methods.

Weaknesses:

-	The authors did not provide any new significant suggestions for any of the three problems. The proposed method is simply a combination of previous works.
-	The experimental results are not significantly strong due to the lack of some baselines and inadequate benchmark problems.

**Questions For Authors:**

1.	What do the authors think about incorporating MSR with other techniques from HDBO baselines? As MSR improve the GP modelling and AF optimization, it can be incorporated with several heuristics such as trust region (TuRBO), adaptive subspace embedding (BAxUS/Bounce), additivity structure (RDUCB). These heuristics for HDBO eventually must deal with the GP fitting and AF optimizing problems, so improving them by MSR techniques can be a great help.

**Relation To Broader Scientific Literature:**

The three key findings are in line with previous results. The authors did quite well to summarize them and illustrate in a comprehensive way.

**Theoretical Claims:**

There is no theoretical claim.

---

> ### Author Rebuttal · Authors · 2025-03-31
>
> We are grateful for the reviewer's thoughtful feedback and recognition of our work's strengths, particularly the systematic summary of high-dimensional Bayesian optimization (HDBO) issues and the effectiveness of our proposed methodology. In response to the constructive comments, we have addressed the concerns and further strengthened our contributions.
>
> **Structure of the Lasso-DNA and Mopta08 benchmarks and inclusion of additional benchmarks**
>
> We recognize the reviewer's concern about the structural risks associated with Lasso-DNA and Mopta08. Since these benchmarks are nearly always included in related work, we included them to demonstrate the effectiveness of our method in scenarios with specific structural properties against other methods. In the spirit of ensuring a broader evaluation, we added the SVM benchmark [1] (https://ibb.co/jkx3MXN1) and will add another benchmark for the camera-ready version at the latest.
>
> **Add baselines that use the RAASP sampling strategy, such as TuRBO, and compare against recent SOTA methods like RDUCB.**
>
> As demonstrated in [2], the DSP method we compare to already significantly outperforms RDUCB. Given this established performance gap, we have focused our comparisons on DSP. However, we acknowledge RDUCB and other milestones in HDBO in the related work section. To provide additional data points, as suggested by the reviewer, we have now added TuRBO (https://ibb.co/jkx3MXN1). TuRBO is competitive on the Mopta08 and Lasso-DNA benchmarks and has good anytime performance on Humanoid for the first half of the optimization but stagnates in the second half.
>
> **Include a baseline that uses RAASP without scaled length scales to isolate the effect of RAASP.**
>
> We have included TuRBO as a baseline that uses RAASP without scaled length scales (https://ibb.co/jkx3MXN1). We further compare MLE without scaled length scales to RAASP (https://ibb.co/6J7g8XRn) and will add this result to the appendix.
>
> **MSR is simply a combination of previous works.**
>
> While MSR builds on existing techniques, its practical significance lies in effectively combining them to address the challenges of HDBO and provide a straightforward yet powerful approach that practitioners can readily adopt.
>
> **Revise Section 4 to identify and describe each method mentioned clearly.**
>
> We will revise Section 4 to improve clarity, ensuring each method is identified and described, making the experimental setup and benchmarks easier to follow.
>
> **Potential for incorporating MSR with other HDBO techniques.**
>
> Integrating MSR with methods such as TRs, adaptive subspace embeddings, and additive structures is a relevant future work direction, which could further enhance the performance and applicability of MSR in various HDBO scenarios. We will add this idea in the future work section.
>
> [1] Eriksson, David, and Martin Jankowiak. "High-dimensional Bayesian optimization with sparse axis-aligned subspaces." Uncertainty in Artificial Intelligence. PMLR, 2021.
>
> [2] Hvarfner, Carl, Erik Orm Hellsten, and Luigi Nardi. "Vanilla Bayesian Optimization Performs Great in High Dimensions." International Conference on Machine Learning.

---

### Decision · Program_Chairs · 2025-05-01

**Decision:**

Accept (poster)

**Comment:**

This paper investigates the problems of Bayesian optimization in high-dimensional space, discussing vanishing gradient and the length-scale prior.  It also proposes mitigations for the problems.

Although there is no theoretical contribution, and the experiments are not comprehensive, all reviewers found that the paper is well-written, well-organized, and provides many impactful insights, which facilitate new research in the ML community.  Minor concerns have been addressed in the rebuttal, and expected to be reflected to the camera-ready version.